# Gas Permeation Property of Silicon Carbide Membranes Synthesized by Counter-Diffusion Chemical Vapor Deposition

**DOI:** 10.3390/membranes10010011

**Published:** 2020-01-06

**Authors:** Takayuki Nagano, Koji Sato, Koichi Kawahara

**Affiliations:** Japan Fine Ceramics Center, 2-4-1, Mutsuno, Atsuta-ku, Nagoya 456-8587, Japan; kosato@jfcc.or.jp (K.S.); kawahara@jfcc.or.jp (K.K.)

**Keywords:** hydrogen, amorphous, silicon carbide, alumina, chemical vapor deposition

## Abstract

An amorphous silicon carbide (SiC) membrane was synthesized by counter-diffusion chemical vapor deposition (CDCVD) using silacyclobutane (SCB) at 788 K. The SiC membrane on a Ni-γ-alumina (Al_2_O_3_) α-coated Al_2_O_3_ porous support possessed a H_2_ permeance of 1.2 × 10^−7^ mol·m^−2^·s^−1^·Pa^−1^ and an excellent H_2_/CO_2_ selectivity of 2600 at 673 K. The intermittent action of H_2_ reaction gas supply and vacuum inside porous support was very effective to supply source gas inside mesoporous intermediate layer. A SiC active layer was formed inside the Ni-γ-Al_2_O_3_ intermediate layer. The thermal expansion coefficient mismatch between the SiC active layer and Ni-γ-Al_2_O_3_-coated α-Al_2_O_3_ porous support was eased by the low decomposition temperature of the SiC source and the membrane structure.

## 1. Introduction

Silicon carbide (SiC) membranes with high strength and high chemical stability can be used under high pressure and corrosive atmospheres at elevated temperatures. The synthesis of SiC membranes has been reported in polymer precursor pyrolysis and chemical vapor deposition (CVD)/chemical vapor infiltration (CVI).

In the polymer precursor pyrolysis, many researchers have reported gas permeation behavior. Polycarbosilane (PCS) is converted to amorphous SiC by the heat treatment at >1073 K. Therefore, PCS has been used as a pre-ceramic polymer. Shelekhin and co-workers synthesized a polymer/inorganic composite membrane on porous Vycor glass by the pyrolysis of PCS at 743 K and reported a hydrogen/nitrogen permeance ratio (H_2_/N_2_) of 14.3 [1]. Kusakabe and co-workers reported a polymer/inorganic composite/γ-Al_2_O_3_/α-Al_2_O_3_ membrane with H_2_ permeance of 4 × 10^−8^ mol·m^−2^·s^−1^·Pa^−1^ and H_2_/N_2_ of 20 at 773 K by the pyrolysis of PCS [2]. Lee and co-workers reported an SiOC membrane with H_2_ permeance of 2.7 × 10^−9^ mol·m^−2^·s^−1^·Pa^−1^ and H_2_/N_2_ of 20 at 473 K by pyrolysis of polydimethylsilane (PMS) [3]. These membranes were cured in oxygen atmosphere to form Si–O–Si by cross-linking into a network polymer. The oxygen introduced in the Si–C network affected the stability during hydrothermal exposure [4,5].

On the other hand, some researchers have reported PCS-derived membranes that did not require an oxygen curing process. Suda and co-workers reported that SiC membranes synthesized using p-diethynylbenzene and Pt_2_ (1,3-divinyltetramethyldisiloxane)_3_ as a catalyst possessed H_2_ permeance of 3 × 10^−8^ mol·m^−2^·s^−1^·Pa^−1^ and H_2_/N_2_ of 150 at 373 K [6,7]. The authors reported that the SiC/γ-Al_2_O_3_/α-Al_2_O_3_ membrane synthesized by PCS without an oxygen curing process possessed a H_2_ permeance of 1.3 × 10^−7^ mol·m^−2^·s^−1^·Pa^−1^ and H_2_/N_2_ of 8.5 at 873 K [8]. The thermal expansion coefficient difference between SiC active layer and γ-Al_2_O_3_/α-Al_2_O_3_ porous support was large. Therefore, the improvement of H_2_/N_2_ was difficult due to the crack formation at the thin SiC active layer. Takeyama and co-workers investigated the synthesis of SiC/γ-Al_2_O_3_/α-Al_2_O_3_ by exposure to electron beam irradiation in helium (He) at room temperature and reported a H_2_ permeance of 3.1 × 10^−7^ mol·m^−2^·s^−1^·Pa^−1^ and H_2_/N_2_ selectivity of 51 at 523 K [9].

In the CVD/CVI process, Takeda and co-workers synthesized that SiC/γ-Al_2_O_3_/α-Al_2_O_3_ membrane by CVI with an alternating gas supply of H_2_ + SiH_2_Cl_2_ and H_2_ + C_2_H_2_ in the temperature range 1073–1173 K [10]. Their membrane possessed a H_2_ permeance of 1.4 × 10^−8^ mol·m^−2^·s^−1^·Pa^−1^ and H_2_/N_2_ of 3.36 at 623 K. Pages and co-workers synthesized SiC membranes inside an asymmetric porous alumina tube by plasma-enhanced CVD. Their permeation mechanism was Knudsen diffusion [11]. Sea and co-workers synthesized a SiC membrane in the macropores of a α–Al_2_O_3_ support tube by CVD using tri-isopropylsilane (TPS) in the temperature range 973–1023 K with a forced cross-flow through the porous wall [12]. The permeation mechanism was Knudsen diffusion in the temperature range 323–673 K. Ciora and co-workers synthesized microporous SiC/γ-Al_2_O_3_/α-Al_2_O_3_ by CVD using 1, 3-disilabutane (DSB) and TPS. The TPS-derived membranes were hydrothermally stable [13]. The authors also tried to improve H_2_/N_2_ selectivity in a PCS-derived SiC/γ-Al_2_O_3_/α-Al_2_O_3_ membrane by chemical vapor infiltration (CVI) using H_2_ + SiH_2_Cl_2_ and H_2_ + C_2_H_2_ [14]. However, the improvement of H_2_ permselectivity was affected by the densification of the Si–C network during surface modification at elevated temperatures.

Generally, inorganic membranes, except zeolite, are composed of an active layer for separating gas, an intermediate layer for suppressing pinhole defects on the active layer, and a porous support. In the preparation of a microporous membrane by CVD and CVI, the gas permeation property strongly depended on the mesoporous intermediate layer structure. The authors synthesized an amorphous silica membrane with an optimized intermediate structure by CDCVD and reported a H_2_/N_2_ permeance ratio of >10,000 [15]. The highest H_2_/N_2_ permeance ratio was obtained at the intermediate layer structure of both the mean pore size of 8 nm and the thickness of 3 μm. Nomura and co-workers compared gas permeation behavior in amorphous silica membranes on γ-Al_2_O_3_/α-Al_2_O_3_ substrates with different pore sizes of 4 nm and 13 nm [16]. They reported that the H_2_ permeance and the H_2_/N_2_ permeance ratio in SiO_2_/γ-Al_2_O_3_(13nm)/α-Al_2_O_3_ membrane were 13/200 and 47/1000 of that in SiO_2_/γ-Al_2_O_3_(4nm)/α-Al_2_O_3_ membrane, respectively. The hydrothermal stability of the γ-Al_2_O_3_ intermediate layer was important for membrane durability [17]. The authors improved the hydrothermal stability of γ-Al_2_O_3_ by 5 mol% Ni doping (Ni-γ-Al_2_O_3_) [18]. α-Al_2_O_3_ porous support is the most popular for inorganic membranes. However, there is a large thermal expansion coefficient difference between Al_2_O_3_ (7.2 × 10^−6^/K) and SiC (3.7 × 10^−7^/K).

Recently, SiC-based porous supports with high gas permeability were developed [19,20,21,22,23,24]. In these porous supports with large pore size, it is difficult to suppress both the decreasing of permeability and the formation of membrane defects due to the penetration of sol-gel solution into large pores. Dabirs and co-workers synthesized an asymmetric porous SiC tube with He permeance of 6 × 10^−5^ mol·m^−2^·s^−1^·Pa^−1^ [25]. However, it was not easy to control the SiC multi-layer structure using polystyrene. Moreover, the commercial supply of SiC tubes was limited and expensive.

In this study, we synthesized a SiC/Ni-γ-Al_2_O_3_/α-Al_2_O_3_ membrane by counter-diffusion chemical vapor deposition (CDCVD) using silacyclobutane (SCB) with a decomposition temperature of 448 K, as shown in Figure 1. The thermal expansion coefficient mismatch can be solved by lowering CVD temperature and minimizing SiC volume inside the Ni-γ-Al_2_O_3_/α-Al_2_O_3_ porous support. The CDCVD method has the advantage of suppressing membrane defects by the membrane structure in which an active layer exists inside the intermediate layer [15].

## 2. Experimental Procedure

α-Al_2_O_3_ porous capillary tube with mean pore size of 150 nm (outer diameter = 2.9 mm, inner diameter = 2.1 mm, length = 400 mm, NOK Corporation, Tokyo, Japan) was used as the substrate. The effective membrane area with a length of 5 cm was located at the center of the substrate. Both ends of the substrate were sealed with glass at 1273 K. The dip-coating solution for γ-Al_2_O_3_ was prepared by reacting 0.05 M of aluminum-tri-sec-butoxide (Al(OCH(CH_3_)C_2_H_5_)_3_ = ATSB, 97%, Sigma-Aldrich, Tokyo, Japan) and 0.1 M isopropyl alcohol (Kanto chemical Co., Inc., Tokyo, Japan) with double-distilled water at 363 K. After the addition of ATSB, the mixture was maintained at 363 K for at least 1 h to evaporate both the isopropyl alcohol and the formed butanol. The mixture was subsequently cooled down to room temperature and peptized with 1 M HNO_3_ at a pH of about 3. During the synthesis, the sol was stirred vigorously. The peptized mixture was refluxed at 363 K for 15 h, yielding a very stable 0.05 M boehmite sol with a clear whitish-blue appearance. Doping of this sol was performed by mixing it with an aqueous solution of Ni(NO_3_)_2_·6H_2_O (Wako Pure Chemical Industries, Ltd., Tokyo, Japan) to enhance the hydrothermal stability. The amount of Ni(NO_3_)_2_·6H_2_O was 5 mol% [18].

The dip-coating solution for Ni-γ-Al_2_O_3_ was obtained by diluting boehmite sol (γ-AlOOH) with a 3.5 wt% solution of polyvinyl alcohol (PVA, mean molecular weight = 72,000: Kanto Kagaku Co., Ltd., Tokyo, Japan) at 363 K. The end of the α-Al_2_O_3_ porous capillary was plugged with silicon cork. The outer surface of the substrate wrapped with Teflon tape was dipped in the solution for 10 s, whereas the inner surface of the capillary was evacuated to obtain a pinhole-free membrane using a rotary pump. After dip coating, the membrane was dried for 2 h in air. It was then calcined at 1123 K for 1 h. The heating/cooling rate was 1 K/min. This dipping–drying–firing sequence was repeated twice.

The pore size distribution of the Ni-γ-Al_2_O_3_ intermediate layer on the α-Al_2_O_3_ porous capillary was analyzed using a nanopermporometer (TNF-3WH-110ME: Seika Sangyo Co., Tokyo, Japan). The noncondensable gas was N_2_, and the liquid used as condensable vapor was water. The specimen was set in the apparatus after annealing at 393 K for 1 h.

The silicon carbide coating on a Ni-γ-Al_2_O_3_-coated α-Al_2_O_3_ substrate was performed by CDCVD, as shown in Figure 2. The substrate was coaxially fixed in a stainless-steel tube and placed in an electric tubular furnace. The temperature of SCB was controlled to 298 K with a mantle heater. SCB was supplied through the outer surface of the substrate by controlling the Ar flow rate at 7.2 sccm. The source gas was diluted with Ar at a flow rate of 64.3 sccm. Hydrogen (H_2_) was introduced into the inner surface of the substrate at the flow rate range 265–300 sccm. The pattern of the source gas supply, reaction gas supply and vacuum inside the tube is shown in Figure 3. The vacuum process of source gas inside the Ni-γ-Al_2_O_3_ intermediate layer has the role of suppressing the formation of membrane defects. The furnace temperature was set to 788 K to avoid powder pollution in the CVD chamber.

The specimen for scanning electron microscopy (SEM) was prepared using a focused ion beam (FIB) system (Model FB-2100, Hitachi Co., Tokyo, Japan) at acceleration voltages of 10–40 kV. The specimen surface was coated with platinum to suppress charge up. The thickness of the platinum layer was 60 nm. Tungsten was deposited on the surface of the sampling area in the FIB system to protect the top layer of the membrane from gallium ion sputtering during FIB milling. The membrane was dug out by FIB milling and small specimens (3 μm × 25 μm × 15 μm) was prepared. The specimen was lifted out using a tungsten needle and transferred to TEM grid. The specimen was then fixed onto the TEM grids by FIB-assisted deposition and thinned by FIB milling.

The top surface and cross-sectional structure of Ni-γ-Al_2_O_3_/α-Al_2_O_3_ capillary was observed by SEM (Model S-8000, Hitachi, Tokyo, Japan). The composition of the top layer was analyzed by energy-dispersive X-ray spectroscopy (EDS) with SEM.

X-ray photoelectron spectroscopy (XPS) analysis was performed using (PHI 1800, ULVAC PHI, Inc., Chigasaki, Japan) to evaluate the SiC membrane structure. The X-ray source was an Al Kα (1253.6 eV) anode kept 350 W. Analyzer pass energy of 23.5 eV was used. The Shirley model was used for establishing the background. The peak shapes were modeled by Gaussian function. A curve fitting analysis was performed by the least-squares method.

Single gas permeance (*P*) was evaluated by a constant-volume manometric method [26]. Permeance data can be measured from 1 × 10^–5^ to 1 × 10^–12^ mol·m^–2^·s^–1^·Pa^–1^ by changing experimental conditions. The membrane was coaxially fixed in a stainless-steel tube and placed in an electric furnace. The sealing portion using o-ring was cooled for water. The outside of the membrane was filled with pure gas under atmospheric pressure conditions. The inside of the membrane was evacuated using a rotary pump. After terminating evacuation, the rate of pressure increase at the inside of the membrane was measured five times to calculate permeance. The average of the third, fourth, and fifth measurements was accepted as permeance data. The gas permeance at each temperature was evaluated in the order of He, H_2_, CO_2_, Ar, and N_2_. Permselectivity was defined as the permeance ratio of the two gases. For example, the H_2_/N_2_ permselectivity is given by the ratio *P*H_2_/*P*N_2_.

## 3. Results and Discussion

### 3.1. Microstructure of the SiC Membrane

The pore size distribution of the Ni-γ-Al_2_O_3_ intermediate layer is shown in Figure 4. The pore sizes of the sample calcined at 1123 K for 1 h were almost distributed between 4 and 8 nm. The average pore size was 5.9 nm.

The cross-sectional backscattered electron image for SiC/Ni-γ-Al_2_O_3_/α-Al_2_O_3_ is shown in Figure 5. The membrane was composed of two layers. The α-Al_2_O_3_ capillary tube was uniformly coated with Ni-γ-Al_2_O_3_ layer and the thickness of the Ni-γ-Al_2_O_3_ layer was approximately 2.3 μm. The contrast in the Ni-γ-Al_2_O_3_ layer originated from the heating history.

The cross-sectional SEM image and EDS line analysis for SiC/Ni-γ-Al_2_O_3_/α-Al_2_O_3_ membrane are shown in Figure 6. The interface between Ni-γ-Al_2_O_3_ and α-Al_2_O_3_ porous support was set to the line corresponding to 3 μm in the x-axis. The surface of the SiC active layer was consistent with the Pt peak in the EDS analysis. The entire top layer was composed of silicon, carbon, aluminum, and oxygen. The content of C was not constant in the Ni-γ-Al_2_O_3_ intermediate layer. The intensity of carbon was high at the reach of the depth of the 0.8 μm from the membrane surface. Therefore, the effective membrane thickness was estimated to be less than 0.8 μm.

Accordingly, the synthesized sample was considered to be composed of two layers, namely, the α-Al_2_O_3_ porous support and the Ni-γ-Al_2_O_3_ intermediate layer gradually modified with SiC.

The results of the XPS analysis for the SiC/Ni-γ-Al_2_O_3_/α-Al_2_O_3_ membrane are shown in Figure 7. The binding energy of C_1s_ showed that this active layer was composed of SiC (283.6 eV) and C (284.4 eV). On the other hand, the binding energy of Si_2P_ showed that the active layer was composed of SiC (100.2 eV), SiO_x_C_y_ (101.4 eV), and SiO_2_ (102.2 eV). The oxygen contamination was thought to come from the source gas or minor leaks. The C/Si atomic ratio for the active layer was 1.05. Therefore, a part of the C–C bond in SCB was thought to have been decomposed by H_2_ during CDCVD.

### 3.2. Gas Permeation Property

Single gas permeance of the Ni-γ-Al_2_O_3_/α-Al_2_O_3_ porous capillary is shown in Figure 8. The experimental data at 673 K were plotted versus the square root of molecular weight. The linear regression fit is presented in Figure 8. The plots show a good linear dependence, confirming that the transport of the gases is induced mainly by Knudsen diffusion. The permeance ratio of H_2_ over the other gases (He: 1.38; CO_2_: 4.47; Ar: 4.09; N_2_: 3.39) showed good agreement with the value of theoretical Knudsen diffusion (He: 1.41; CO_2_: 4.67; Ar: 4.45; N_2_: 3.73), indicating that the pores are controlled finely and homogeneously [27].

The reaction gas flow rate dependence on single gas permeance of SiC membranes at 673 K is shown in Figure 9. The H_2_ flow rate had a large effect on the permeance of CO_2_, Ar, and N_2_. In the CDCVD method, the balance of the gas pressure difference between the source gas and the reaction gas is important to plug the pores in the intermediate layer. In the membrane synthesized at a H_2_ flow rate of 300 sccm, the membrane defect was thought to be formed by the lack of source gas inside the intermediate layer.

The deposition time dependence on the single gas permeance of SiC membranes at 673 K is shown in Figure 10. The increase in deposition time led to an increase in membrane thickness and the plugging of pores. At the deposition time of 9 min, the permeance of CO_2_, Ar, and N_2_ was almost the same. Hydrogen permeance and H_2_/N_2_ permselectivity were well-balanced at the deposition time of 9 min. In the CDCVD process, the reaction between the source gas and the reaction gas was automatically stopped after the pore plugging in the intermediate layer. We believe that the vacuum process of the source gas must be effective for pore plugging.

Gas permeation properties through the SiC/-γ-Al_2_O_3_/α-Al_2_O_3_ membrane in the temperature range 323–673 K are shown in Figure 11. After CDCVD modification, He and H_2_ permeance tended to increase with a decreasing kinetic diameter. However, CO_2_, Ar, and N_2_ permeance was almost the same in the temperature range 323–673 K. The He, H_2_, CO_2_, Ar, and N_2_ permeance at 673 K was 1.7 × 10^–7^, 1.2 × 10^–7^, 4.8 × 10^–11^, 5.1 × 10^–11^, and 4.0 × 10^–11^ mol·m^–2^·s^–1^·Pa^–1^, respectively. The H_2_ permeance at 673 K was three orders of magnitude higher than the CO_2_ permeance at 673 K, and the H_2_/CO_2_ permselectivity was calculated to be 2600. This membrane showed molecular sieving behavior. The pore size distribution of the SiC active layer was very sharp and average pore size was considered to be about 0.3 nm between the H_2_ and CO_2_ kinetic diameters.

The temperature dependence on each gas permeance for SiC/-γ-Al_2_O_3_/α-Al_2_O_3_ membrane is shown in Figure 12. The permeance for He and H_2_ increased with increasing permeation temperature. Accordingly, the dominant permeation mechanism for He and H_2_ was considered to be activated diffusion [28,29].

The activation energy of permeation was obtained by fitting the gas permeance data to the Arrhenius expression:Q = Q_0_ exp (−Ea/RT)(1)
where Q is the permeance, Q_0_ is the pre-exponential factor (mol·m^–2^·s^–1^·Pa^–1^), Ea is the activation energy (J·mol^–1^), R is the gas constant (8.314 J·mol^–1^·K^–1^), and T is the temperature (K).

The apparent activation energy for the He and H_2_ permeance in silicon carbide membrane was 9.8 and 11.2 kJ/mol in the temperature range from 323 to 673 K, respectively. He and H_2_ through the specimen of SiC/Ni-γ-Al_2_O_3_/α-Al_2_O_3_ mainly permeated through the amorphous network in the temperature range 323–673 K. The activation energy of He and H_2_ was close to that of the SiOC membranes, as shown in Table 1. These values were thought to reflect Si–O–C bond of the SiC active layer and the low calcination temperature in CDCVD. On the other hand, these values were 1.0, 1.2, and 0.1 kJ/mol for CO_2_, Ar, and N_2_ permeance, respectively, at the temperature range 323–673 K. These data show that the membrane defect was small in this sample. The authors report that the heating history during calcination and membrane thickness had the effect of lowering of the activation energy in amorphous SiC membranes [8].

Therefore, the CDCVD process using a precursor with Si–C bond at a low decomposition temperature is very effective for synthesizing a SiC membrane with a sharp pore size distribution.

In further work, the development of SiC mesoporous intermediate layer/SiC porous support with high permeability is very important to increase total durability of SiC membrane.

## 4. Conclusions

Amorphous silicon carbide membranes with a high H_2_/CO_2_ permeance ratio of 2600 were successfully synthesized on Ni-γ-Al_2_O_3_-coated α-Al_2_O_3_ by counter-diffusion chemical vapor deposition. The dominant permeation mechanism for He and H_2_ was activated diffusion. The mesoporous pores became plugged with the discrete SiC active layer in the intermediate layer, and the use of SCB with a low decomposition temperature was effective in suppressing the thermal expansion coefficient mismatch between the SiC active layer and Ni-γ-Al_2_O_3_/α-Al_2_O_3_ porous capillary.

## Figures and Tables

**Figure 1 membranes-10-00011-f001:**
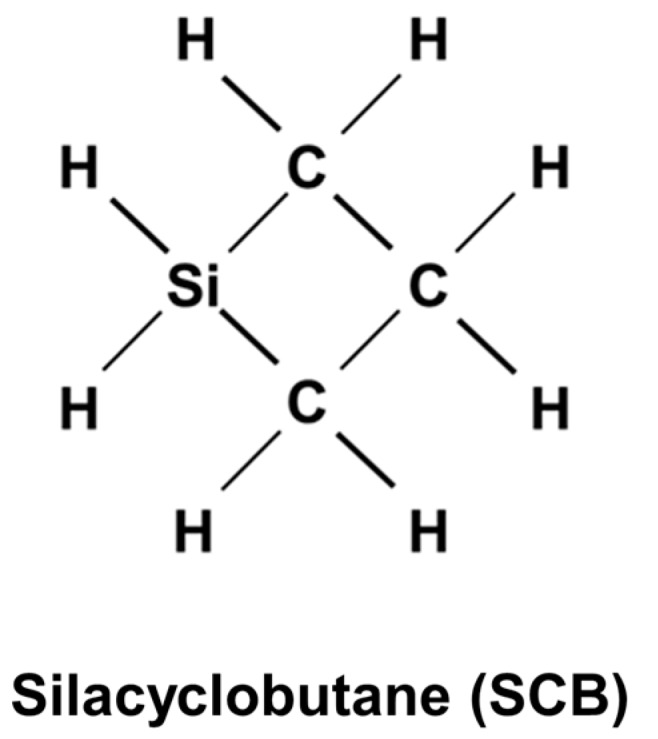
Chemical vapor deposition (CVD) source for SiC membranes.

**Figure 2 membranes-10-00011-f002:**
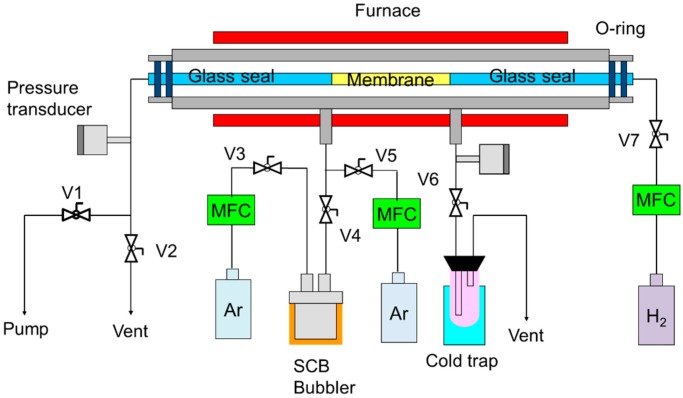
Counter-diffusion chemical vapor deposition apparatus for SiC membranes.

**Figure 3 membranes-10-00011-f003:**
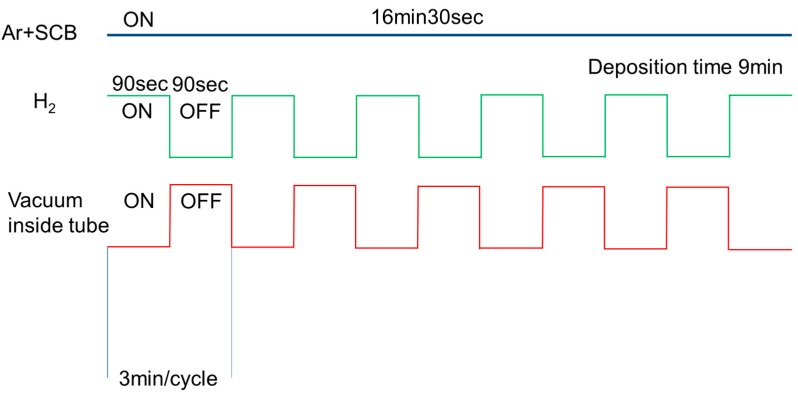
Pattern of source gas supply, reaction gas supply, and vacuum inside tube.

**Figure 4 membranes-10-00011-f004:**
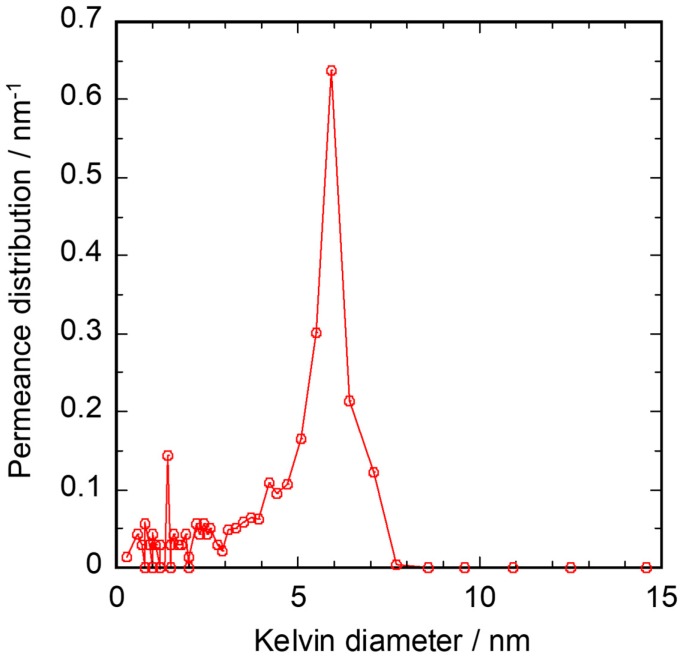
Pore size distribution of Ni-γ-Al_2_O_3_/α-Al_2_O_3_ porous capillary.

**Figure 5 membranes-10-00011-f005:**
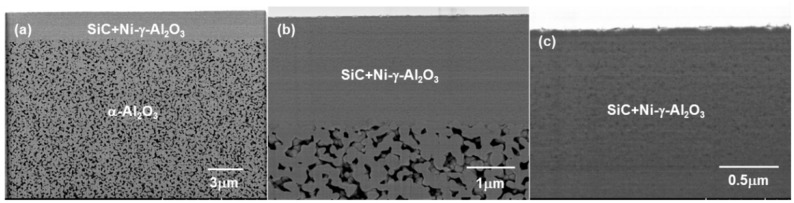
Cross-sectional backscattered electron images of SiC/Ni-γ-Al_2_O_3_/α-Al_2_O_3_ membrane: (**a**) ×5000, (**b**) ×20,000, and (**c**) ×50,000.

**Figure 6 membranes-10-00011-f006:**
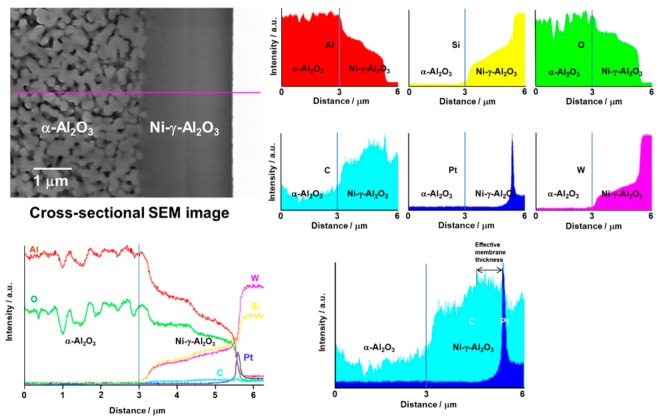
SEM image and EDS line analysis for SiC/Ni-γ-Al_2_O_3_/α-Al_2_O_3_ membrane.

**Figure 7 membranes-10-00011-f007:**
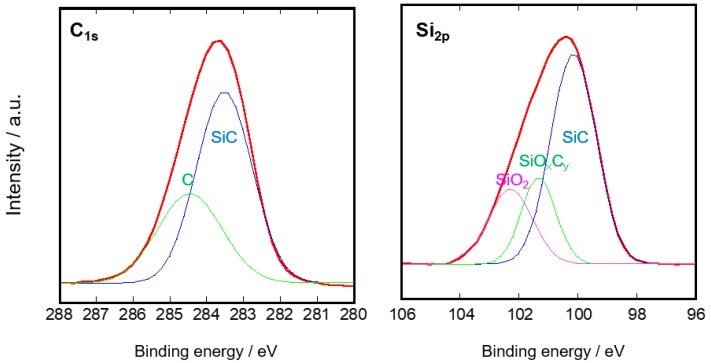
Result of XPS analysis for SiC/Ni-γ-Al_2_O_3_/α-Al_2_O_3_ membrane.

**Figure 8 membranes-10-00011-f008:**
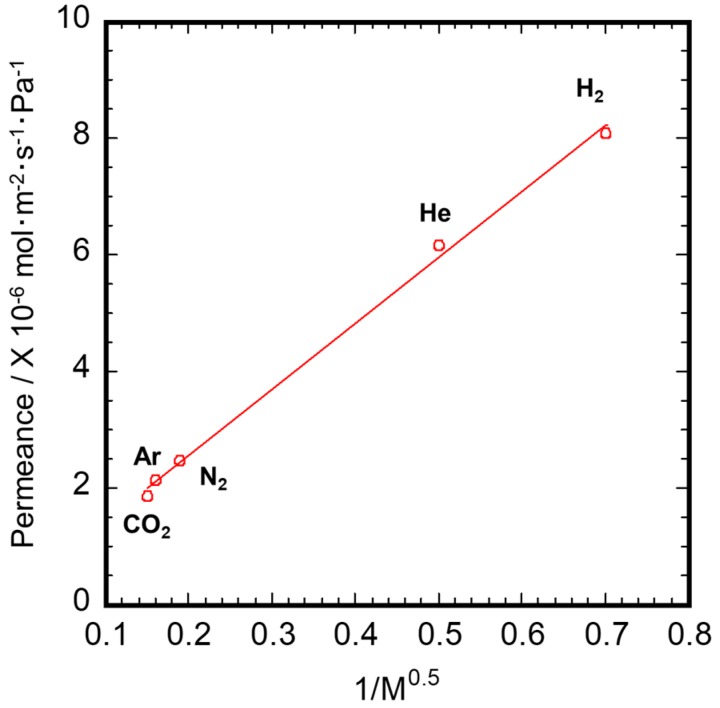
Single gas permeance of Ni-γ-Al_2_O_3_/α-Al_2_O_3_ porous capillary.

**Figure 9 membranes-10-00011-f009:**
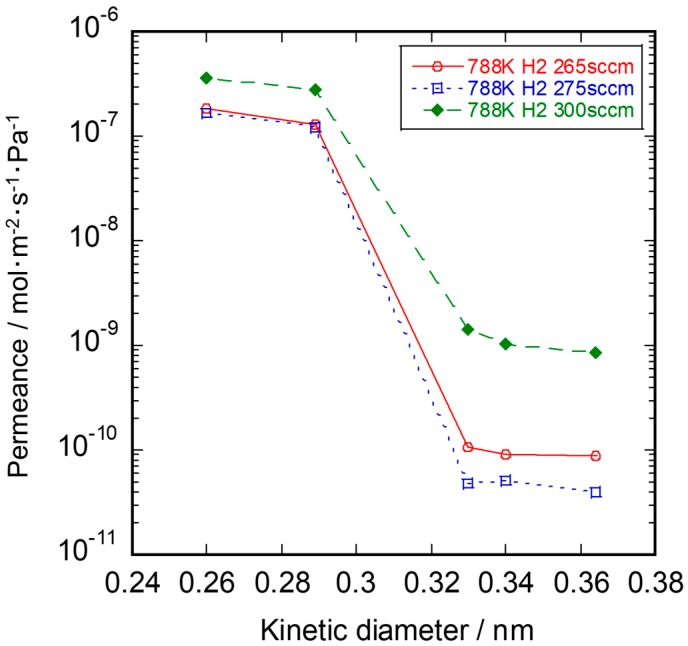
Reaction gas flow rate dependence on single gas permeance of SiC membranes.

**Figure 10 membranes-10-00011-f010:**
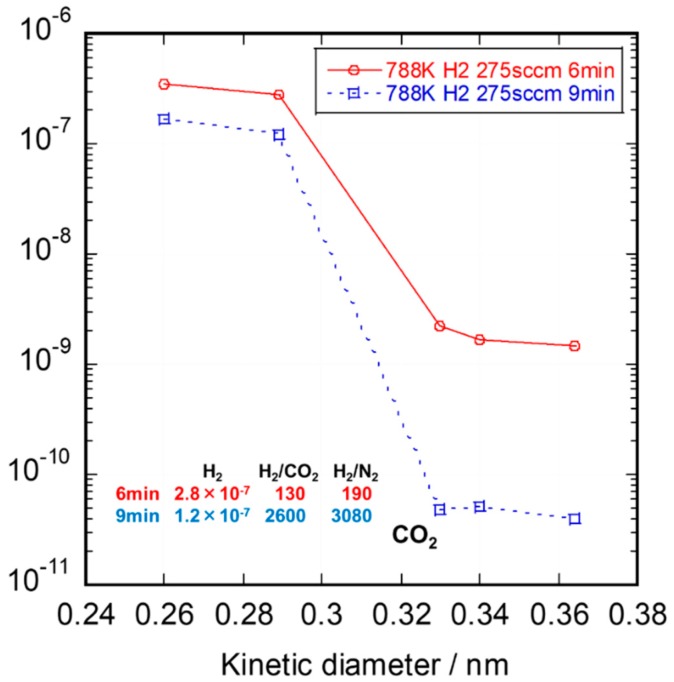
Deposition time dependence on single gas permeance of SiC membranes.

**Figure 11 membranes-10-00011-f011:**
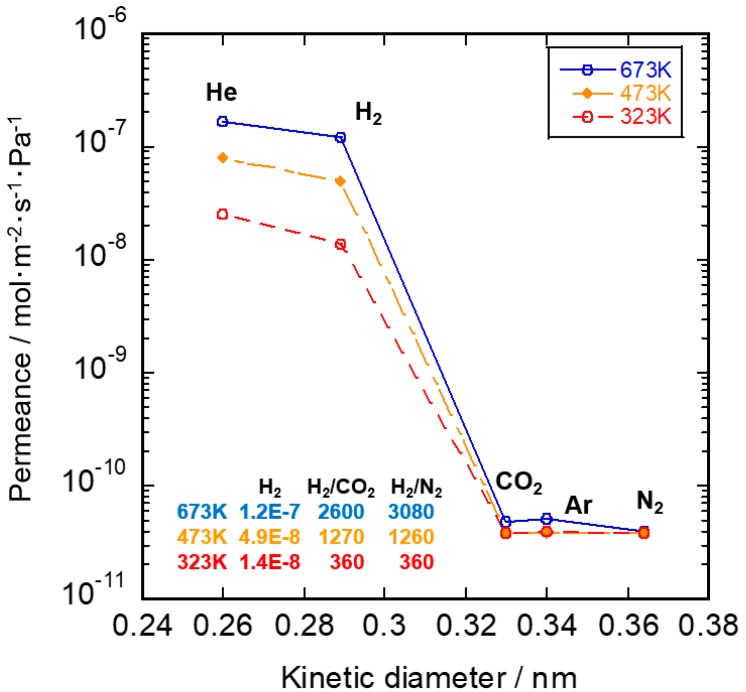
Single gas permeance of SiC membrane in the temperature range 323–673 K.

**Figure 12 membranes-10-00011-f012:**
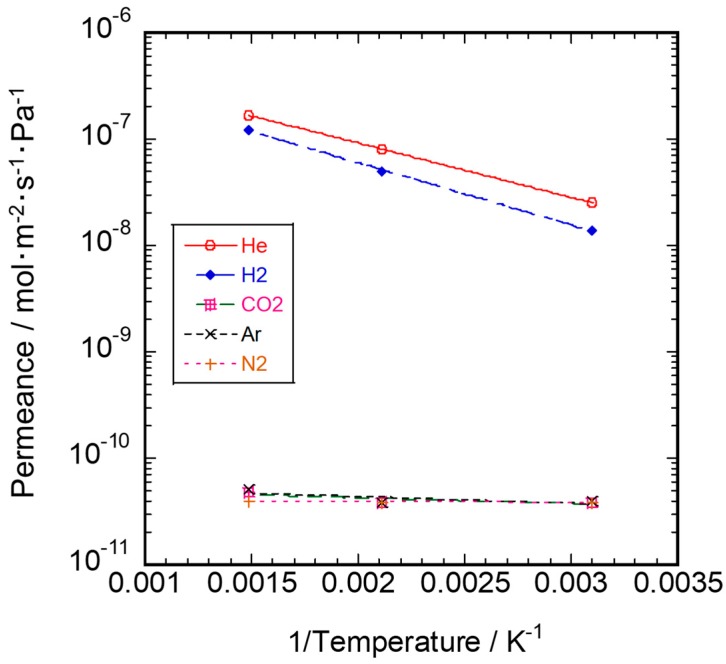
Temperature dependence of single gas permeance for SiC membrane.

**Table 1 membranes-10-00011-t001:** Apparent activation energy for He and H_2_ in SiOC, SiC, and SiO_2_ membranes.

Membrane	Permeate Temp. (K)	Activation Energy (kJ/mol)		Ref.
		He	H_2_	
SiOC	283–773	7–13	7–15	[4]
SiOC	350–473	9.8–15.4	16.3–16.7	[3]
SiC	773–873	9.4	6.0	[8]
	573–773	2.7	1.2	[8]
	573–873	1.2–1.9	0.4–1.0	[8]
(PCS+CVI)	573–773	0.6	0.09	[14]
	773–873	1.2	0.3	[14]
SiO_2_	573–873	14	15.3	[30]
SiO_2_	373–873	9.8	14.8	[31]
SiO_2_	373–873	8.1	16.8	[15]
This study	323–673	9.8	11.2	-

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
