# Peer review of "Gas Permeation Property of Silicon Carbide Membranes Synthesized by Counter-Diffusion Chemical Vapor Deposition"

_membranes, 2020, doi:10.3390/membranes10010011_

Round 1
Reviewer 1 Report
The manuscript is devoted to develop novel membranes by counter diffusion chemical vapor deposition and investigate them in gas separation.
The authors present technique of membrane preparation and methods of characterization.
However, the experimental data are not reported with exhaustive explanations and the manuscript should be consider after a major revision:
From the introduction it is not quite clear what advantages the proposed method of obtaining SiC membrane have. Or this is the only one? Could authors ad any references?
Authors are strongly recommended to add more modern references on the topic in the Introduction.
It is not appropriate to use such form “E-7” in the scientific paper. Please, correct it throughout the text (×10-7 or ·10-7).
It was not evident to me from EDS analysis why there is no Ni signs on the line. Also, it would be pertinent to present SEM images of SiC layer and information about it thickness. Please add magnitude and scale for SEM image.
Could authors mention how they confirmed mass transfer mechanism? It is unclear why firstly Knudsen diffusion is confirmed and then it changes to activated diffusion and molecular sieving. Please clarify the statements with an explanation and references.
There is no Table 1 in the manuscript.
In the part with deposition time dependence there is a great difference between the permeances for CO2, Ar, N2 for 6 min and 9 min while the permeances for H2 and He are slightly different. Please give some assumptions about this phenomenon.
Why do not authors give information on SiC layer porosity in Section 3.1 along with an intermediate layer. Was it determined by the same method? How did it affect permeability? Was it important only at high temperature?
The authors should give the comparison of the prepared material with reported membrane in the literature or commercially available membranes for H2/CO2 and H2/N2 permeability and selectivity.
Author Response
Thank you very much for providing important insights. In following sections, you will find our responses to each of your points and suggestions. We are grateful for the time and energy you expended on our behalf.
Comment1
From the introduction it is not quite clear what advantages the proposed method of obtaining SiC membrane have. Or this is the only one? Could authors ass any references?
Answer1
Counter-diffusion chemical vapor deposition method has been applied for silica-based membranes. The reaction gases for oxide material were O2 and O3. In our experiment, we selected H2 as reaction gas. As far as we know, this is the first report applied for SiC membrane.
Comment2
Authors are strongly recommended to add more modern references on the topic in the introduction.
Answer2
We performed literature search on SiC membrane, again. However, we could not catch the point. Are you able to introduce a literature specifically?
Comment3
It is not appropriate to use such form “E-7” in the scientific paper. Please, correct it throughout the text (x10-7 or ï½¥10-7).
Answer3
We asked to rewrite our manuscript for native researcher.
Comment4
It has not evident to me from EDS analysis why there is no Ni sighns on the line. Also, it would be pertinent to present SEM images of SiC layer and information about it thickness. Please add magnitude and scale for SEM image.
Answer4
We edited EDS result as shown in Fig. 5. The information about scale and the position of SiC/Ni-g-Al2O3 interface is added. The thickness of Ni-g-Al2O3 layer is shown in text. The addition of Ni content was 5 mol%. NiO was not detected by XRD as shown in another literature [17]. The intensity of carbon was high at the reach of 0.8 mm from membrane surface. Therefore, we estimated that the effective membrane thickness was less than 0.8 mm.
Comment5
Could authors mention how they confirmed mass transfer mechanism? It is unclear why firstly Knudsen diffusion is confirmed and then it changes to activated diffusion and molecular sieving. Please clarify the statements with an explanation and references.
Answer5
We added the single gas permeance of Ni-g-Al2O3/a-Al2O3 capillary tube in Fig. 7. The experimental data were plotted versus the square root of molecular weight. The plots showed a good linear dependence.
Comment6
There is no Table 1 in the manuscript.
Answer6
We have submitted the manuscript contained with Table I. However, we could not confirm Table I in preprint manuscript. We will again submit the manuscript with Table 1.
Comment7
In the part with deposition time dependence there is a great difference between the permeances for CO2, Ar, N2 for 6 min and 9 min while the permeances for H2 and He are slightly different. Please give some assumptions about this phenomenon.
Answer7
In counter-diffusion chemical vapor deposition process, the reaction between the source gas and the reaction gas was automatically stopped after pore plugging in the intermediate layer. We believe that the vacuum process of the source gas must by effective for pore plugging.
Comment8
Why do not authors give information on SiC layer porosity in Section 3.1 along with an intermediate layer. Was it determined by the same method? How did it affect permeability? Was it important only at high temperature?
Answer8
We measured the pore size distribution of Ni-g-Al2O3 intermediate layer by the principle on capillary condensation of a vapor and its ability to block the permeation of a non-condensable gas. In this method, we can directly obtain the pore size distribution of >0.5 nm. Therefore, it was difficult to measure the pore size of SiC active layer of 0.3 nm.
Comment9
The authors should give the comparison of the prepared material with reported membrane in the literature or commercially available membranes for H2/CO2 and H2/N2 permeabiliry and selectivity.
Answer9
The permeance and selectivity data in SiC membranes were introduced in introduction. These data were not high in comparison of Pd, SiO2, carbon, and polymeric membranes. We expect the future development of SiC membranes.
Again, thank you for giving us the opportunity to strengthen our manuscript with your valuable comments and queries. We have worked hard to incorporate your feedback and hope that these revisions persuade you to accept our submission.

Reviewer 2 Report
The authors report an Si-C based inorganic membrane for high temperature gas separation. The membrane was fabricated with CVD process and the mismatched thermal expension co-efficiency between the film and substrate was successfully addressed. The author also tried to explain the mechanism of gas transport. Overall, I find this manuscript is in a good quality and it can be accepted for publication as it is. There is one just minor comments: more experimental details are welcomed to explain the membrane separation setup for the separation process.
Author Response
Thank you very much for providing important insights. In following sections, you will find our responses to each of your points and suggestions. We are grateful for the time and energy you expended on our behalf.
Comment1
More experimental detailed are welcomed to explain the membrane separation setup for the separation process.
Answer1
We added membrane separation setup in experimental procedure and the single gas permeance of Ni-g-Al2O3/a-Al2O3 capillary tube in Fig. 7. The experimental data were plotted versus the square root of molecular weight. The plots showed a good linear dependence.
We also asked to rewrite manuscript for native researcher.
Again, thank you for giving us the opportunity to strengthen our manuscript with your valuable comments and queries. We have worked hard to incorporate your feedback and hope that these revisions persuade you to accept our submission.

Reviewer 3 Report
In this manuscript, the authors synthesized SiC/Ni-r-Al2O3/a-Al2O3 membrane using CVD method and characterized it by SEM, EDS, XPS. The influence of H2 flow rate, deposition time, and deposition temperature on the gas permeance was studied, respectively. I can’t recommend this manuscript to be accepted at this stage because of the following reasons:
The characterization section is too sketchy to reflect the structure of the membrane. The authors have to re-make Figure 5. A lot of important information is missing, such as scale bar in the SEM image, names of all the X-axis and Y-axis, what the curves represent, etc. The figure caption is not informative at all, which should introduce every subfigure.
To analyze XPS data in Figure 6, the authors should perform curve fitting instead of simply compare the binding energy range among different bonds.
The authors should keep consistent with the name of the membrane. They used SiC/N-r-Al2O3/a- Al2O3 mostly, but also used SiC/Ni-g-Al2O3/a-Al2O3 (Figure 6 caption), or SiC/r-Al2O3/a-Al2O3 (Line 203). Besides, in some places they used r-Al2O3 instead of Ni-r-Al2O3, such as Figure 5 SEM image and Line 165.
Author Response
Thank you very much for providing important insights. In following sections, you will find our responses to each of your points and suggestions. We are grateful for the time and energy you expended on our behalf.
Commemt1
The characterization section is too sketchy to reflect the structure of the membrane. The authors have to re-make figure 5. A lot of important information is missing, such as scale bar in the SEM image, names of all the X-axis and Y-axis, what the curves represent, etc. The figure caption is not informative at all, which should introduce every subfigure.
Answer1
EDS data were remake to distinguish each element and SiC/Ni-g-Al2O3 interface. The position of interface, scale in SEM image, names of the X-axis and Y-axis were added.
Comment2
To analyze XPS data in Fig. 6, the authors should perform curve fitting instead of simply compare the binding energy range among different bonds.
Answer2
XPS data were replotted and fitted with gaussian curves. The data of SiC powder were eliminated to enlarge fitting curves.
Comment3
The authors should keep consistent with the name of the membrane. They used SiC/Ni-g-Al2O3/a-Al2O3 mostly, but also used SiC/Ni-g-Al2O3/a-Al2O3 (Line 203). Besides, in some places they used g-Al2O3 instead of Ni-g-Al2O3, such as a Fig. 5 SEM image and Line 165.
Answer3
The notation of intermediate layer was unified to Ni-g-Al2O3.
Again, thank you for giving us the opportunity to strengthen our manuscript with your valuable comments and queries. We have worked hard to incorporate your feedback and hope that these revisions persuade you to accept our submission.

Reviewer 4 Report
This manuscript described preparation of defect-less SiC membrane under lowering CVD temperature to reduce thermal expansion mismatch between SiC active layer and the support. I think experiments were carefully performed, and presented data supported their proposition. I have minor comments in the manuscript.
1. There are missing words (Ni- - alumina-coated …etc ) in Abstract, probably coming from errors during formatting.
2. In pp 3, line 85, correct PH to pH
3. In pp 3, line 102, did you mean a stainless steel tube??
4. In pp 6, line 163, “Al2O3” should be modified into its subscript forms.
5. For consistency of whole manuscript, please modify helium to He when it describes He permeance.
6. There is a typo in pp 8 of line 202. (Herium to He)
Author Response
Thank you very much for providing important insights. In following sections, you will find our responses to each of your points and suggestions. We are grateful for the time and energy you expended on our behalf.
Comment1
There at missing words (Ni-g-Al2O3-coated ・・・ etc) in Abstract, probably coming from errors during formatting.
Answer1
After preprint process, symbol font in our manuscript was garbled.
Comment2
In pp 3, line 85, correct PH to pH
Answer2
We corrected PH to pH.
Comment3
In pp6, line 163, “Al2O3” should be modified into its subscript forms.
Answer3
We corrected Al2O3 to Al2O3.
Comment4
For consistency pf whole manuscript, please modify helium to He when it describes He permeace.
Answer4
We corrected helium permeance to He permeace.
Again, thank you for giving us the opportunity to strengthen our manuscript with your valuable comments and queries. We have worked hard to incorporate your feedback and hope that these revisions persuade you to accept our submission.

Round 2
Reviewer 1 Report
Question 1 and 2 (previous review). This is an unacceptable answer. The lack of articles in this area indicates the irrelevance of the problem. Different solutions to the problem of separation of gas mixtures by the membrane method already exist. Why should specialists pay attention to your work? What are its distinctive features and advantages over classical polymer or metal membranes, with which a hydrogen / nitrogen or hydrogen/helium mixture is separated?
As for the articles, I was able to find a few with similar keywords in a short time.
DOI: 10.1016/j.memsci.2019.117464
DOI: 10.1021/ie503626u
DOI: 10.1016/j.jeurceramsoc.2016.06.048
DOI: 10.5004/dwt.2019.23713
I am sure that the authors can significantly improve the Introduction by making extra efforts.
Question 4 (previous review). Authors haven’t answered the question: «Also, it would be pertinent to present SEM images of SiC layer»
Question 5 (previous review). «The linear regression fit is presented in Fig. 7. The plots show a good linear dependence, confirming that the transport of the gases is induced mainly by Knudsen diffusion.» line 179
«Accordingly, the dominant permeation mechanism for He and H2 was considered to be activated diffusion» line 207
Please add confirming references
Question 9 (previous review). «The permeance and selectivity data in SiC membranes were introduced in introduction. These data were not high in comparison of Pd, SiO2, carbon, and polymeric membranes. We expect the future development of SiC membranes.»
It is worth adding a similar wording about the main result and future plans in the Conclusion.
Author Response
Thank you very much for providing important insights. In following sections, you will find our responses to each of your points and suggestions. We are grateful for the time and energy you expended on our behalf.
Comment1 and 2
This is an unacceptable answer. The lack of articles in this area indicates the irrelevance of the problem. Different solutions to the problem of separation of gas mixture by the membrane method already exist. Why should specialists pay attention to your work? What at its distinctive features and advantages over classical polymer or metal membranes, with which a hydrogen/nitrogen or hydrogen/helium mixture is separated?
As for the articles, I was able to find a few with similar keywords in a short time. I am sure that the authors can significantly improve the introduction by making extra efforts.
Answer
Our SiC membrane shows molecular sieving behavior. We believe that the gas permeation behavior should be discussed the ideal separation factor. The biggest feature of ceramic membranes is excellent heat-resistance. Therefore, SiC active layer/SiC mesoporous intermediate layer/SiC porous support structure is expected. Unfortunately, we have some problems to solve.
The references on SiC porous supports with high permeability were added in introduction. The pore sizes for these supports are too large to suppress membrane defect in sol-gel coating. We have to develop new method to form SiC mesoporous intermediate layer on SiC porous support with high permeability.
Comment4
Authors haven’t answered the question:<<Also, it would be pertinent to present SEM images of SiC layer>>
Answer
We added the cross-sectional backscattered electron images for SiC/Ni-g-Al2O3/a-Al2O3 membrane in Fig. 5.
Comment5
<<The linear regression fit is presented in Fig. 7. The plots show a good linear dependence, confirming that the transport of the gases is induced mainly Knudsen diffusion.>> line 179
<<Accordingly, the dominant permeation mechanism for He and H2 was considered to be activated diffusion>> line 207
Please add confirming references
Answer
We added references.
Comment9
<<The permeance and selectivity data in SiC membranes were introduced in introduction. These data were not high in comparison of Pd, SiO2, carbon, and polymeric membranes. We expected the future development of SiC membranes.
It is worth adding a similar wording about the main result and future plans in the Conclusion.
Answer
We added the references about SiC-based porous supports with high permeability. These supports have a potential to increase gas permeance, drastically. However, the large pore sizes in these supports are not suitable for sol-gel coating of mesoporous intermediate layer. We believe the improvement of sol-gel-coating method to achieve remarkable progress.
Again, thank you for giving us the opportunity to strengthen our manuscript with your valuable comments and queries.

Round 3
Reviewer 1 Report
In the present form the manuscript can be published in Polymers